# Insoluble Dietary Fiber from Soybean Residue (Okara) Exerts Anti-Obesity Effects by Promoting Hepatic Mitochondrial Fatty Acid Oxidation

**DOI:** 10.3390/foods12102081

**Published:** 2023-05-22

**Authors:** Jiarui Zhang, Sainan Wang, Junyao Wang, Wenhao Liu, Hao Gong, Zhao Zhang, Bo Lyu, Hansong Yu

**Affiliations:** 1College of Food Science and Engineering, Jilin Agricultural University, Changchun 130118, China; 2Division of Soybean Processing, Soybean Research & Development Center, Chinese Agricultural Research System, Changchun 130118, China; 3Sinoglory Health Food Co., Ltd., Liaocheng 252000, China

**Keywords:** insoluble dietary fiber, okara, obesity, fatty acid oxidation, regulatory mechanism

## Abstract

Numerous investigations have shown that insoluble dietary fiber (IDF) has a potentially positive effect on obesity due to a high-fat diet (HFD). Our previous findings based on proteomic data revealed that high-purity IDF from soybean residue (okara) (HPSIDF) prevented obesity by regulating hepatic fatty acid synthesis and degradation pathways, while its intervention mechanism is uncharted. Consequently, the goal of this work is to find out the potential regulatory mechanisms of HPSIDF on hepatic fatty acid oxidation by determining changes in fatty acid oxidation-related enzymes in mitochondria and peroxisomes, the production of oxidation intermediates and final products, the composition and content of fatty acids, and the expression levels of fatty acid oxidation-related proteins in mice fed with HFD. We found that supplementation with HPSIDF significantly ameliorated body weight gain, fat accumulation, dyslipidemia, and hepatic steatosis caused by HFD. Importantly, HPSIDF intervention promotes medium- and long-chain fatty acid oxidation in hepatic mitochondria by improving the contents of acyl-coenzyme A oxidase 1 (ACOX1), malonyl coenzyme A (Malonyl CoA), acetyl coenzyme A synthase (ACS), acetyl coenzyme A carboxylase (ACC), and carnitine palmitoyl transferase-1 (CPT-1). Moreover, HPSIDF effectively regulated the expression levels of proteins involved with hepatic fatty acid β-oxidation. Our study indicated that HPSIDF treatment prevents obesity by promoting hepatic mitochondrial fatty acid oxidation.

## 1. Introduction

Obesity is becoming a major public health issue around the world, with its prevalence rising year after year. Obesity can cause severe consequences, such as type 2 diabetes, metabolic syndrome, cardiovascular disease, and neurodegenerative diseases [1,2,3]. HFD is thought to increase the risk of obesity by increasing lipid synthesis, decreasing fatty acid oxidation, and impairing triglyceride (TG) export. At present, the primary methods for treating obesity are dieting, drug treatment, and surgical treatment, all of which have varying degrees of side effects. In consequence, there is a desperate demand for secure and effective ways to prevent and manage obesity.

The most active tissue for fatty acid oxidation is the liver. The oxidation of fatty acids is classified as β-oxidation and particular oxidation modes such as α-oxidation and ω-oxidation, with β-oxidation being the primary oxidation pathway [4,5,6]. β-oxidation occurs in the mitochondria and peroxisomes. AMP-activated protein kinase (AMPK) is a key signaling element that regulates hepatic fatty acid oxidation. When AMPK is activated, it inhibits fat accumulation and promotes fatty acid oxidation by regulating the enzymatic activities of sterol regulatory element-binding protein-1c (SREBP-1c) and peroxisome proliferator-activated receptor-α (PPARα) [7,8,9,10]. Silent mating type information regulation 2 homolog-1 (SIRT1) increases the transcriptional activity of PPARα mainly through proliferator-activated receptor gamma coactivator-1α (PGC-1α) deacetylation, which in turn promotes the β-oxidation of fatty acids in the liver. In addition, some proteins situated downstream of the AMPK pathway, such as CPT-1, are intimately related to β-oxidation. CPT-1 is localized on the outer mitochondrial surface and reduces the intracellular fatty acid concentration by catalyzing the beta-oxidation of fatty acids [11,12,13]. Therefore, obesity can be prevented and controlled by promoting fatty acid oxidation.

Soybean residue (okara) is the insoluble portion remaining after filtering the water-soluble portion during soymilk or soybean curd (tofu) production. Although large amounts of okara are yielded by the food industry, most of it is wasted because the high moisture content. As we all know, okara contains a variety of nutrients, especially dietary fiber (DF), which is regarded as the seventh nutrient. DF is essential for maintaining human health and is classified into soluble dietary fiber and IDF according to its water solubility. Numerous forward-looking studies have demonstrated the critical role of IDF in obesity prevention. Frank et al. discovered that insoluble cereal fiber supplementation significantly lowered weight gain and improved insulin sensitivity compared to long-term supplementation with soluble cereal fiber [14]. Another study showed that IDF from *Pleurotus eryngii* has a preventive effect on obesity through its modulation of the gut microbiota [15]. Our previous study demonstrated that high-purity IDF from okara (HPSIDF) plays a beneficial role in preventing obesity by regulating hepatic fatty acid synthesis and degradation pathways in HFD-fed mice. However, the mechanism of intervention is unclear [16].

Therefore, according to the results of previous research, this work was conducted to further explore the regulatory mechanisms of HPSIDF on fatty acid oxidation by analyzing the changes in the content of mitochondrial and peroxisomal oxidation-related enzymes, the production of oxidative intermediates and final products, fatty acid content and composition, and the expression levels of hepatic fatty acid oxidation-associated proteins in HFD induced mice.

## 2. Materials and Methods

### 2.1. Materials and Reagents

Crude soybean residue with 60% IDF content was purchased from Shandong Sinoglory Health Food Co., Ltd. (Liaocheng, China). The CDF was used to prepare the HPSIDF by the enzymatic method. The specific conditions are as follows: α-amylase at 95 °C for 35 min, neutral protease at 60 °C for 30 min, and amyloglucosidase at 60 °C for 30 min [17]. Kits for total cholesterol (TC), TG, low-density lipoprotein cholesterol (LDL-C), and high-density lipoprotein cholesterol (HDL-C) were derived from the Nanjing Jiancheng Bioengineering Institute (Nanjing, China). ELISA kits for free fatty acids (FFA), 3-hydroxybutyric acid (3-OHB), acetoacetate (ACAC), CPT1, ACOX1, ACS, Malonyl CoA, ACC, hydrogen peroxide (H_2_O_2_), acetyl-coenzyme A (A-CoA), citrate synthase (CS), and succinyl-coenzyme A (SCoA) were derived from Shanghai Enzyme-linked Biotechnology Co., Ltd. (Shanghai, China). HPLC-grade methyl tert-butyl ether (MTBE), methanol (MeOH), and *n*-hexane were derived from Merck (Darmstadt, Germany). Sodium chloride and phosphate were acquired from Sigma-Aldrich (St. Louis, MO, USA). A methanol solution of 15% boron trifluoride was bought from RHAWN (Shanghai, China). The antibodies to PPARα, fatty acid synthase (FAS), and long-chain acyl-coenzyme A dehydrogenase (ACADL) were derived from Wuhan Sanying Biotechnology Co., Ltd. (Wuhan, China). SREBP-1c, AMPK, CPT1, SIRT1, phosphorylated adenylate-activated protein kinase (pAMPK), PGC-1α, ACOX1, long-chain acyl-coenzyme A synthetase (ACSL), and β-actin were provided by Abcam (Cambridge, UK).

### 2.2. Animal Experimental Design

The research was carried out in accordance with the Laboratory Animals Guidelines of Jilin Agricultural University and authorized by the Jilin Agricultural University Laboratory Animal Welfare and Ethics Committee (no. 2019 04 10 005). Six-week-old male C57BL/6J mice (*n* = 40; 18–22 g) were obtained from the Experimental Animal Center of Jilin Agricultural University. Mice were kept in a temperature-controlled (20–25 °C) environment with a 12-h cycle of light/darkness.

Following the first week of acclimatized feeding, mice were separated into four groups for 18 weeks: the normal diet (ND) group, the HFD group, the HFD supplemented with HPSIDF group (HPSIDF, 1000 mg/kg [16]), and the HFD supplemented with L-carnitine group (PC, 40 mg/kg). Notably, L-carnitine has been shown to promote fatty acid metabolism in vivo, which is consistent with the pathway explored in this study, so we chose it as a positive control [18,19]. The body weight and food intake of mice in each group were recorded weekly and daily, respectively. At the end of the trial, mice were anesthetized, and blood was collected through the orbital venous plexus. Then all the animals were euthanized by carbon dioxide. The blood samples were followed by centrifugation at 3000 rpm for 15 min at 4 °C to obtain serum. The liver and fat tissues were obtained by dissection after the execution of mice, weighed, and immediately stored at −80 °C for further analysis.

### 2.3. Biochemical Analysis

The levels of TC, TG, LDL-C, and HDL-C in serum were determined using commercial assay kits. TC and TG levels in the liver were determined using the same kit as the serum assay. Moreover, the levels of FFA, 3-OHB, and ACAC in serum were determined using the ELISA kits.

### 2.4. Analysis of Enzyme Contents Related to Fatty Acid Oxidation

An amount of liver was mixed with saline at 1:9 and homogenized on ice. Then it was centrifugated at 3000 rpm for 15 min at 4 °C to obtain the supernatant. The levels of CPT-1, ACOX1, Malonyl Coenzyme A, ACC, ACS, A-CoA, CS, SCoA, and H_2_O_2_ were analyzed by ELISA kits.

### 2.5. Histological Investigation

Liver and epididymal fat were fixed in 4% paraformaldehyde, engrained in paraffin for making sections (5 μm thickness), and stained using hematoxylin and eosin (H&E).

### 2.6. Analysis of Hepatic Fatty Acid Composition and Content

The liver samples from each group of mice were thawed, and the samples (0.05 g) were mixed with 150 µL of MeOH, 200 μL of MTBE, and 50 μL of 36% phosphoric acid/water (precooled at −20 °C). The mixture was vortexed for 3 min at 2500 rpm and centrifuged at 12,000 rpm for 5 min at 4 °C. Then, 200 μL of supernatant was collected into a new centrifuge tube, blow dry, and 300 μL of a methanol solution of 15% boron trifluoride was added. The mixture was vortexed for 3 min at 2500 rpm and keep in the oven at 60 °C for 30 min. Then 500 μL of *n*-hexane and 200 μL of saturated sodium chloride solution were added accurately at room temperature. After the mixture was vortexed for 3 min and centrifuged at 12,000 rpm and 4 °C for 5 min, 100 μL of *n*-hexane layer solution was transferred for further GC-MS analysis.

A GC-EI-MS system was used to analyze the sample derivates. (GC, Agilent 8890ˈ https://Agilent.com.cn/ (accessed on 13 March 2019); MS, 5977B System, https://Agilent.com.cn/ (accessed on 13 March 2019). The following were the analytical conditions: GC column, DB-5MS capillary column (30 m × 0.25 mm × 0.25 μm, Agilent); Carrier gas, highly pure argon gas (purity > 99.999%); The heating procedure was started at 40 °C (2 min), then increased at 30 °C/min to 200 °C (1 min), 10 °C/min to 240 °C (1 min), and 5 °C/min to 285 °C (3 min); traffic: 1.0 mL/min; inlet temperature: 230 °C; injection volume: 1.0 μL. EI-MS: Agilent 8890-5977B GC-MS System, Temperature: 230 °C; ionization voltage: 70eV; power transmission temperature: 240 °C; four-stage rod temp.: 150 °C; solvent postpone: 4 min; scanning method: SIM.

Qualitative and quantitative analysis. Standard solutions of different concentrations of 0.01, 0.02, 0.05, 0.1, 0.2, 0.5, 1, 2, 5, 10, 20, and 50 μg/mL were prepared, and the peak intensity data corresponding to the different concentrations of the standards were obtained to plot the standard curves. The integrated peak areas of the detected liver samples were brought to the standard curve to calculate the concentrations, which were then further calculated to give absolute content data for the different substances in the actual samples.

The fatty acid content of the sample (μg/g) = c∗V3/1000∗V1/V2/m

Meaning of the letters in the formula:

c: Concentration values obtained by substituting the integrated peak area of the sample into the standard curve (μg/mL);

V1: Volume of sample extraction solution (μL);

V2: Volume of collected supernatant (μL);

V3: Volume of resolution (μL);

m: The sample weight (g).

### 2.7. Western Blot Analysis

The liver was homogenized in RIPA lysis solution, centrifuged at a speed of 12,000 rpm at 4 °C for 20 min, and the supernatant was collected. The BCA kit was used to determine the content of protein, followed by routine protein blotting analysis. Briefly, protein lysates were isolated by SDS-PAGE and moved to polyvinylidene fluoride (PVDF) films. The blotting film was closed with 5% defatted skim for 1.5 h and then hatched with the corresponding elementary antibodies FAS, SREBP-1c, ACOX1, PPARα, AMPK, pAMPK, SIRT1, PGC-1α, ACADL, ACSL, and CPT1 for the whole night at 4 °C, then incubated with the relevant secondary antibodies for 1.5 h. The bands were analyzed on an iBright CL1000 imaging system using an enhanced chemiluminescence reagent (Invitrogen, Singapore). β-actin was used as a reference to standardize protein expression.

### 2.8. Statistical Analysis

SPSS 19.0 (SPSS, Chicago, IL, USA) was used for statistical analysis. A one-way analysis of variance (ANOVA) and Duncan’s multiple range tests were used to assess the statistical significance, with *p* < 0.05 regarded as statistically significant.

## 3. Results

### 3.1. Effects of HPSIDF on Body Weight, Food Intake and Fat Accumulation in HFD-Fed Mice

As shown in Table 1, there was no remarkable difference in the initial body weight of the mice in each group. After 18 weeks of feeding, the body weight gain of mice was remarkably increased in the HFD group as opposed to the ND group (*p* < 0.05). In comparison to the HFD group, there was a decrease in body weight gain in the HPSIDF and PC groups, especially the HPSIDF group (*p* < 0.05). There was a significant 15% increase in body weight in the HFD group compared to the ND group at the end of the experiment. Compared to the HFD group, weight loss was 15% and 8% in the HPSIDF group and PC group, respectively. In terms of food intake, only the HFD group had a significantly higher food intake in contrast to the other groups (*p* < 0.05), with no significant variation between the other three groups. In addition, we found that chronic HFD caused abnormal accumulation of perirenal, subcutaneous, and epididymal fat compared to ND (*p* < 0.0001) (Figure 1A–C). However, HPSIDF treatment significantly reduced the weight of perirenal, subcutaneous, and epididymal fat induced by HFD (*p* < 0.0001), which was better than the PC group. As shown in Figure 1D, we also found significant adipocyte hypertrophy in HFD-fed mice. Interestingly, HPSIDF treatment considerably alleviated HFD-induced adipocyte hypertrophy compared to the PC group.

### 3.2. Effects of HPSIDF on Serum Biochemical Indicators in HFD-Fed Mice

As shown in Figure 2A–E, serum TC (*p* < 0.0001), TG (*p* < 0.01), LDL-C (*p* < 0.0001), and FFA (*p* < 0.05) contents were memorably elevated in the HFD group in contrast to the ND group. HPSIDF intervention effectively reduced the serum TC (*p* < 0.01), TG (*p* < 0.0001), LDL-C (*p* < 0.0001), and FFA (*p* < 0.001) contents of HFD-fed mice, and there was no considerable difference when compared to the PC group. Furthermore, the HPSIDF and PC groups significantly increased serum HDL-C levels in mice fed with HFD (all *p* < 0.0001). As shown in Figure 2F,G, the levels of 3-OHB (*p* < 0.0001) and ACAC (*p* < 0.05) in the HFD group were observably higher than those of the ND group, and these alterations were effectively reversed by the HPSIDF intervention (*p* < 0.05, *p* < 0.01). Moreover, ACAC levels were dramatically reduced in the PC group as opposed to the HFD group (*p* < 0.0001).

### 3.3. Effects of HPSIDF on Hepatic Steatosis in HFD-Fed Mice

The chronic HFD resulted in significantly higher liver TC (*p* < 0.001), TG (*p* < 0.0001), and FFA (*p* < 0.0001) levels than ND (Figure 3A). As expected, the concentrations of TC, TG, and FFA were remarkably reduced in the HPSIDF (*p* < 0.01, *p* < 0.001, *p* < 0.001) and PC groups as opposed to the HFD group (*p* < 0.05, *p* < 0.01, *p* < 0.01). In Figure 3B, increased lipid droplet accumulation with marked steatosis was observed in HFD-induced mice. The degree of steatosis in the hepatocytes was ameliorated in both the HPSIDF and PC groups, with red staining of the hepatocyte cytoplasm and reduced lipid droplets.

### 3.4. Effects of HPSIDF on the Activity of Enzymes Related to Hepatic Fatty Acid Oxidation and the Production of Intermediate and Final Products in HFD-Fed Mice

We found that ACOX1 (*p* < 0.0001), malonyl CoA (*p* < 0.01), ACS (*p* < 0.01), and ACC (*p* < 0.01) levels were observably enhanced in HFD-fed mice compared to ND-fed mice (Figure 4). However, the levels of ACOX1, malonyl CoA, and ACC were markedly reduced in the HPSIDF (*p* < 0.01, *p* < 0.0001, *p* < 0.01) and PC (*p* < 0.05, *p* < 0.01, *p* < 0.001) groups. Interestingly, CPT-1 and ACS were dramatically increased in the HPSIDF (*p* < 0.01, *p* < 0.001) and PC (*p* < 0.05, *p* < 0.001) groups as opposed to the HFD group. Furthermore, HPSIDF treatment notably promoted the production of SCoA (*p* < 0.05), CS (*p* < 0.01), and A-CoA (*p* < 0.001) and inhibited the synthesis of H_2_O_2_ (*p* < 0.0001) in the liver of HFD-fed mice.

### 3.5. Effects of HPSIDF on Hepatic Fatty Acid Content and Composition in HFD-Fed Mice

As shown in Figure 5, long term HFD leads to the accumulation of hexanoic acid (C6:0), octanoic acid (C8:0), decanoic acid (C10:0), palmitic acid (C16:0), stearic acid (C18:0), behenic acid (C22:0), tricosanoic acid (C23:0), lignoceric acid (C24:0), cis-10-pentadecenoic acid (C15-1), cis-10-pentadecenoic acid (C18-1n9c), trans-9-octadecenoic acid (C18-1n9t), cis-8,11,14-eicosatrienoic acid (C20-3n6), cis-7,10,13,16,19-docosapentaenoic acid (DPA) (C22-5), cis-4,7,10,13,16,19-docosahexaenoic acid (C22-6n3) and nervonic acid (C24-1) in the liver. After HPSIDF intervention, the contents of hexanoic acid (C6:0), octanoic acid (C8:0), decanoic acid (C10:0), lauric acid (C12:0), myristic acid (C14:0), palmitic acid (C16:0), heptadecanoic acid (C17:0), stearic acid (C18:0), tricosanoic acid (C23:0), lignoceric acid (C24:0), myristoleic acid (C14-1), cis-10-pentadecenoic acid (C15-1), cis-10-heptadecanoic acid (C17-1), linoleic acid (C18-2n6c), γ-linolenic acid (C18-3n6), cis-11-eicosenoic acid (C20-1), trans-11-eicosenoic acid (C20-1T), cis-11,14-eicosadienoic acid (C20-2), cis-11,14,17-eicosatrienoic acid (C20-3n3), cis-8,11,14-eicosatrienoic acid (C20-3n6), cis-5,8,11,14,17-eicosapentaenoic acid(EPA) (C20-5n3), arachidonic acid (C20-4n6), trans-13-docosenoic acid (C22-1T), cis-13,16-docosadienoic acid (C22-2), cis-7,10,13,16,19-docosapentaenoic acid (DPA) (C22-5), nervonic acid (C24-1) were memorably reduced in HFD-fed mice. Thus, the HPSIDF intervention significantly promoted the oxidation of medium- and long-chain fatty acids in HFD-fed mice.

### 3.6. Effects of HPSIDF on the Expression Levels of Hepatic Fatty Acid β-Oxidation-Associated Proteins in HFD-Fed Mice

As shown in Figure 6A, several key enzymes for fatty acid β-oxidation were determined in the liver. Long-term HFD markedly upregulated the expression levels of ACSL (*p* < 0.01), ACOX1 (*p* < 0.0001), and SREBP-1c (*p* < 0.001), while downregulating the levels of PPARα (*p* < 0.05) and ACADL (*p* < 0.05) in contrast to the ND group. However, these changes were reversed by HPSIDF intervention, except for the ACSL levels, which were downregulated (*p* < 0.0001). Moreover, HPSIDF treatment observably upregulated the expression levels of CPT-1 (*p* < 0.0001), PGC-1α (*p* < 0.0001), SIRT1 (*p* < 0.05), and pAMPK (*p* < 0.001), and downregulated the expression levels of FAS (*p* < 0.05) in contrast to the HFD group.

## 4. Discussion

The majority of nutrition studies over the last few years have emphasized the value of a high fiber intake. The World Health Organization, Food and Agriculture Organization (WHO/FAO), and the European Food Safety Authority (EFSA) recommend dietary fiber intake of not less than 25 g/day [20,21]. Although most of the suggested beneficial effects of fiber intake are attributed to the viscosity of SDF, results from forward-looking cohort studies concordantly suggest that the intake of IDF, but not SDF, is strongly linked to a decreased risk of obesity and overweight [22,23,24]. Therefore, based on the recommended intake we investigated the effect of different doses of HPSIDF on lipid metabolism in HFD-fed mice in the previous study. It was found that the high-dose group had better effects compared to the low and medium-dose groups. Thus, we chose the high-dose group to explore the effect of HPSIDF on high-fat diet-induced fatty acid oxidation. It is well known that the main feature of obesity is an excessive amount of fat, which leads to a variety of complications, such as dyslipidemia, liver steatosis, insulin resistance, and inflammation [25,26,27]. In this study, HPSIDF intervention remarkably improved fat accumulation, dyslipidemia, and hepatic steatosis in HFD-fed mice. This is in line with the views of Frank et al., who found that supplementation with suitably fermentable insoluble grain fiber prevented HFD-induced obesity and related metabolic disorders [14]. Furthermore, abnormal fat oxidation can raise ketone body levels above the established cutoff, which would then result in ketosis. Interestingly, supplementing with HPSIDF reversed the HFD-induced spike in ketone body levels and brought them back to normal. On the basis of these results, we further explored the potential mechanism of HPSIDF for obesity prevention by studying fatty acid oxidation-related enzymes and proteins.

The liver is an important site of fatty acid metabolism. Huang et al. reported that fatty acids are generated in a series of enzymatic processes and can be oxidized and metabolized to carbon dioxide and water under an adequate oxygen supply. This process is accompanied by the release of tremendous energy. β-oxidation is the primary type of fatty acid oxidation [28]. Kim et al. reported the existence of two different β-oxidation systems, mitochondria and peroxisomes, in mammals as well as higher animals, including humans [29]. Mitochondria mainly oxidize short- and medium-length chain fatty acids, while peroxisomes oxidize substrates such as very-long-chain fatty acids and long-chain fatty acids [30,31]. Research has shown that the peroxisome plays only a minor role in the oxidation of long-chain fatty acids, most of which are oxidized in the mitochondria [32,33]. In addition, the oxidation of fatty acids in the peroxisome is partial and just a carbon chain-shortening reaction. Very long-chain fatty acids enter the peroxisome to form shorter acyl coenzymes, then undergo β-oxidation in the mitochondria to produce acetyl CoA, and finally enter the tricarboxylic acid cycle to produce carbon dioxide and water [34,35]. Therefore, we closely monitored the effects of HPSIDF intervention on the content of enzymes related to fatty acid oxidation and metabolites during oxidation in the liver of HFD-fed mice. Notably, intake of HPSIDF markedly reduced the levels of ACOX1, Malonyl CoA and ACC and increased the levels of CPT-1 and ACS in the liver of HFD-fed mice. Meanwhile, HPSIDF treatment observably promoted the production of SCoA, CS and A-CoA, inhibited the synthesis of H_2_O_2_ and diminished the oxidation of peroxisomal fatty acids. It is reported that excessive peroxisomal fatty acid oxidation inhibits mitochondrial fatty acid oxidation, leading to impaired fatty acid oxidation and metabolic disturbances [21]. We speculate that HPSIDF intervention promotes the rate of mitochondrial oxidation. Chen et al. discovered that chronic consumption of high-erucic acid rapeseed oil caused hepatic steatosis in both animals and humans. This may be due to the increased production of hepatic malonyl coenzyme A in rats by peroxides of erucic acid, which inhibit fatty acid oxidation in mitochondria [36]. Therefore, the levels of malonyl CoA and ACC in the liver were measured in this study. Long-term HFD feeding resulted in a significant increase in malonyl CoA and ACC in the liver. The intervention of HPSIDF clearly reversed this situation. These results demonstrated that HPSIDF promotes overall fatty acid oxidation by promoting mitochondrial fatty acid oxidation and offers to a theoretical foundation for further investigation of its mechanism of action at the protein level.

Studies have shown that hepatic fatty acid metabolism is regulated by the AMPK signaling pathway [37,38]. AMPK contains three subunits: -α, -β, and -γ, among which the -α subunit contains a catalytic phosphorylation site at its NH2 terminal (Thr172) [27,39]. Activation of AMPK phosphorylation downregulates the expression of SREBP-1c to inhibit lipid synthesis while stimulating PPARα expression to promote fatty acid oxidation [40]. Meanwhile, the activation of PPARα leads to increased expression of ACSL1 to provide more conjugated acyl coenzymes for use as fuel via the fatty acid oxidation pathway [41]. Gao et al. also explored the conflicting reports on ACSL, concluding that the destiny of long-chain acyl-coenzyme A in cells is based on the positioning of ACSL1 [42]. In this study, we discovered that long-term HFD resulted in a remarkable increase in the expression levels of SREBP-1C, FAS, and ACOX1. Interestingly, the HPSIDF treatment reversed these changes while significantly upregulating the expression levels of pAMPK, PPARα, CPT-1, ACADL, and ACSL. Our findings indicated that HFD inhibits mitochondrial fatty acid oxidation. This is consistent with our previous hypothesis that HPSIDF intervention could effectively promote mitochondrial fatty acid oxidation. In addition to AMPK activation, research suggests that SIRT1 may also be involved in the regulation of PPARα. According to Cohen et al., fasting and dietary restriction activated SIRT1 and PPARα [43]. Furthermore, PGC-1α is a key coactivator of PPARα signaling and a direct substrate of SIRT1 [44,45]. Aparna et al. demonstrated that SIRT1 regulated PPARα signaling mainly through the activation of PGC-1α and that increased levels of SIRT1 stimulated PPARα activity [46,47,48,49,50,51]. We also determined the protein expression of PGC-1α and SIRT1 in the liver [52,53,54,55]. There was no significant difference in the HFD group in contrast to the ND group, and the HPSIDF intervention greatly upregulated the expression levels of both proteins. Furthermore, ACSL1 and CPT-1 are key rate-limiting enzymes that catalyze mitochondrial fatty acid oxidation. Briefly, ACSL1 catalyzes the production of lipid acyl-coenzyme A at the outer mitochondrial membrane, which is subsequently transported to the mitochondrial matrix by CPT1 to complete the fatty acid oxidation process. ACADL catalyzes the dehydrogenation of long-chain fatty acyl-coenzyme A in the first step of β-oxidation in mitochondria. The HPSIDF intervention significantly upregulated the expression levels of the above key enzymes, which also provided a reasonable explanation for the decrease in the content of medium- and long-chain fatty acids represented by lauric acid (C12:0), myristic acid (C14:0), palmitic acid (C16:0), heptadecanoic acid (C17:0), stearic acid (C18:0), and tridecanoic acid (C23:0) in the HPSIDF group. These results suggest that HPSIDF intervention promotes hepatic fatty acid β-oxidation in HFD-fed mice by activating AMPK phosphorylation to upregulate the expression levels of SIRT1, PGC-1α, and PPARα, while stimulating the expression of downstream proteins such as CPT-1, ACOX1, ACADL, and ACSL.

## 5. Conclusions

In summary, our results indicated that HPSIDF supplementation effectively alleviated body weight gain, fat accumulation, dyslipidemia, and hepatic steatosis induced by HFD. Furthermore, HPSIDF treatment promoted medium- and long-chain fatty acid oxidation in hepatic mitochondria by increasing the contents of CPT-1 and ACS and inhibiting the synthesis of ACOX1, malonyl CoA, and ACC. Meanwhile, HPSIDF treatment dramatically regulated the expression levels of hepatic fatty acid oxidation-related proteins. Overall, our findings gave new perspectives to elucidate the intervention mechanisms of HPSIDF on obesity and play an active role in promoting the comprehensive utilization rate and added value of okara.

## Figures and Tables

**Figure 1 foods-12-02081-f001:**
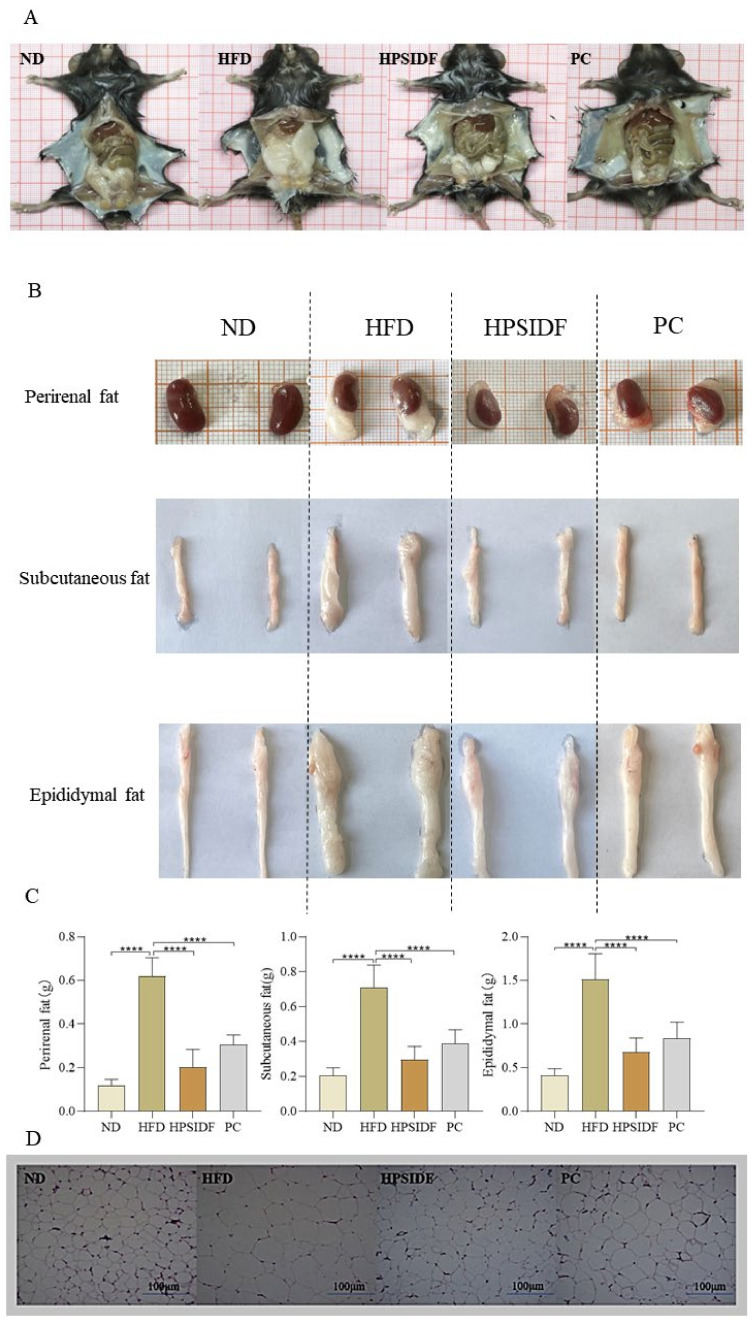
Effects of HPSIDF on fat accumulation in HFD−fed mice. (**A**) Intraperitoneal morphology of mice in each group. (**B**,**C**) Apparent morphology and weight of white adipose tissues. (**D**) Representative H&E staining images of epididymal fat tissue. ND—normal diet-fed group; HFD—high-fat diet−fed group; HPSIDF—high-fat diet plus HPSIDF (1000 mg/kg) fed group; PC—high-fat diet plus L-carnitine (40 mg/kg) fed group. In contrast to the HFD group, **** *p* < 0.0001.

**Figure 2 foods-12-02081-f002:**
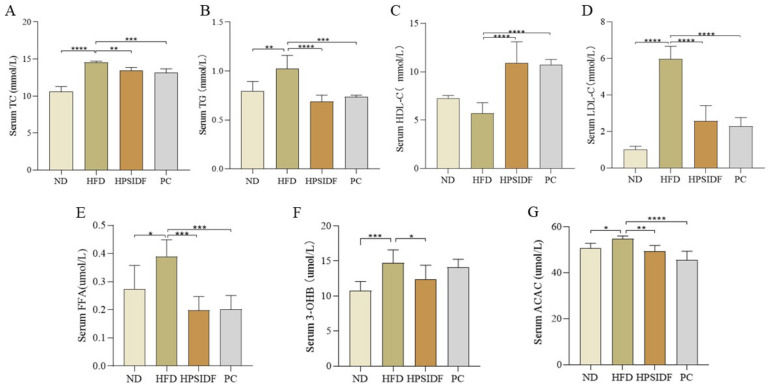
Effects of HPSIDF on serum biochemical factors in HFD−fed mice. (**A**–**E**) Serum TC, TG, HDL-C, LDL-C, and FFA contents. (**F**,**G**) Levels of 3-OHB and ACAC. ND—normal diet−fed group; HFD—high−fat diet−fed group; HPSIDF—high-fat diet plus HPSIDF (1000 mg/kg) fed group; PC—high−fat diet plus L−carnitine (40 mg/kg) fed group. Compared with the HFD group, * *p* < 0.05, ** *p* < 0.01, *** *p* < 0.001 and **** *p* < 0.0001.

**Figure 3 foods-12-02081-f003:**
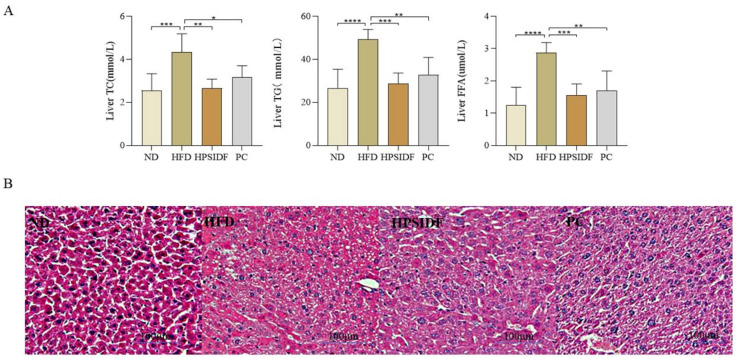
Effects of HPSIDF on hepatic steatosis in mice fed with HFD. (**A**) Levels of hepatic TC, TG and FFA. (**B**) H&E staining of the liver. ND—normal diet−fed group; HFD—high-fat diet−fed group; HPSIDF—high-fat diet plus HPSIDF (1000 mg/kg) fed group; PC—high−fat diet plus L−carnitine (40 mg/kg) fed group. Compared with the HFD group, * *p* < 0.05, ** *p* < 0.01, *** *p* < 0.001, and **** *p* < 0.0001.

**Figure 4 foods-12-02081-f004:**
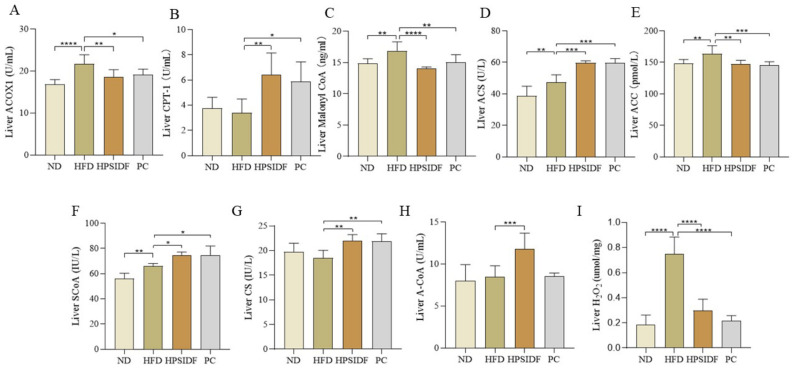
Effects of HPSIDF on the activity of enzymes related to hepatic fatty acid oxidation and the production of intermediate and final products in HFD−fed mice. Determination of ACOX1, CPT-1, Malonyl CoA, ACS, ACC, ScoA, CS, A-CoA, and H_2_O_2_ in the liver (**A**–**I**). ND—normal diet-fed group; HFD—high-fat diet−fed group; HPSIDF—high-fat diet plus HPSIDF (1000 mg/kg) fed group; PC—high-fat diet plus L−carnitine (40 mg/kg) fed group. In contrast to the HFD group, * *p* < 0.05, ** *p* < 0.01, *** *p* < 0.001, and **** *p* < 0.0001.

**Figure 5 foods-12-02081-f005:**
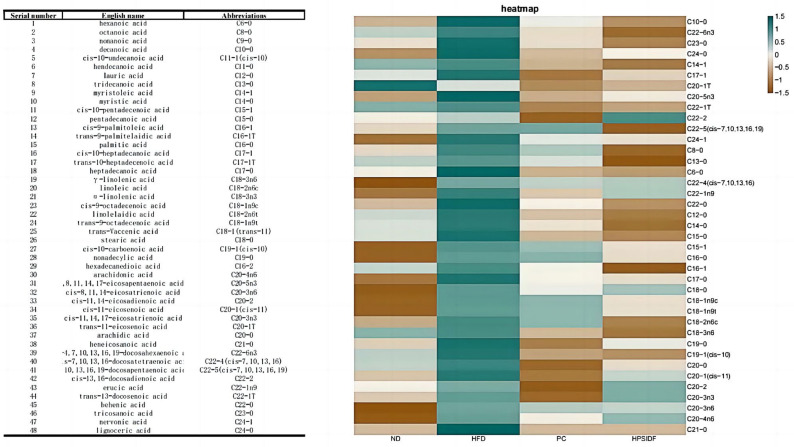
Effects of HPSIDF on hepatic fatty acid composition and content in HFD−fed mice. ND—normal diet−fed group; HFD—high-fat diet−fed group; HPSIDF—high-fat diet plus HPSIDF (1000 mg/kg) fed group; PC—high-fat diet plus L-carnitine (40 mg/kg) fed group.

**Figure 6 foods-12-02081-f006:**
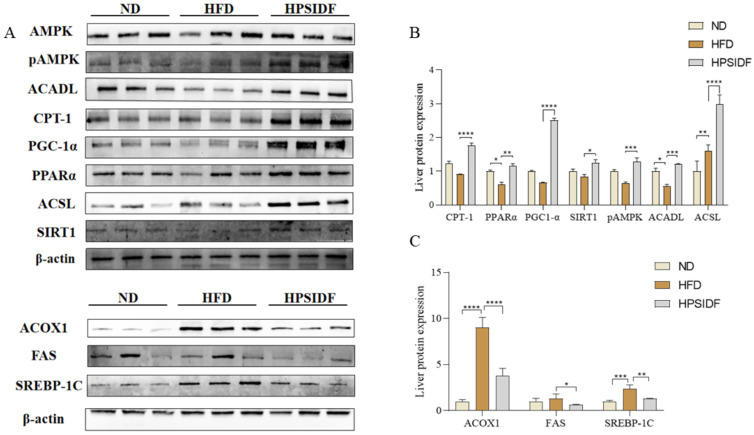
Effects of HPSIDF on the expression levels of hepatic fatty acid oxidation-associated proteins in HFD-fed mice. The levels of CPT-1, PPARα, PGC-1α, SIRT1, pAMPK, ACADL, ACSL, ACOX1, FAS, and SREBP-1c were determined using western blot analysis. The comparative intensities of these protein stripes were analyzed using ImageJ software. β-actin was used as a reference. (**A**) Protein-blotted bands measured. (**B**) CPT-1, PPARα, PGC-1α, SIRT1, pAMPK, ACADL, ACSL protein expression analysis. (**C**) Analysis of ACOX1, FAS, and SREBP-1c protein expression. Values are shown as averages ± SD (*n* = 10). ND—normal diet-fed group; HFD—high-fat diet-fed group; HPSIDF—high-fat diet plus HPSIDF (1000 mg/kg) fed group. In contrast to the HFD group, * *p* < 0.05, ** *p* < 0. 01, *** *p* < 0.001, and **** *p* < 0.0001.

**Table 1 foods-12-02081-t001:** The effect of HPSIDF on body weight and food intake in HFD−fed mice.

Items	ND	HFD	HPSIDF	PC
Initial body weight (g)	20.02 ± 0.93	20.80 ± 1.19	21.10 ± 1.04	21.45 ± 0.63
Final body weight (g)	27.49 ± 2.09 ^b^	31.78 ± 2.77 ^a^	26.90 ± 2.79 ^b^	28.96 ± 2.24 ^b^
Body weight gain (g)	7.47 ± 1.40 ^b^	10.98 ± 2.94 ^a^	5.99 ± 2.40 ^c^	7.50 ± 2.57 ^b^
Food intake (g)	2.13 ± 0.10 ^b^	2.47 ± 0.04 ^a^	2.08 ± 0.07 ^b^	2.15 ± 0.13 ^b^

ND—normal diet-fed group; HFD—high-fat diet-fed group; HPSIDF—high-fat diet plus HPSIDF (1000 mg/kg) fed group; PC—high-fat diet plus L-carnitine (40 mg/kg) fed group. The same column with different letters (a, b, c) are markedly different (*p* < 0.05). Results are shown as average ± SD (*n* = 10).

## Data Availability

Raw data can be provided by the corresponding author upon request.

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
