# Peer review of "Insoluble Dietary Fiber from Soybean Residue (Okara) Exerts Anti-Obesity Effects by Promoting Hepatic Mitochondrial Fatty Acid Oxidation"

_foods, 2023, doi:10.3390/foods12102081_

Round 1

Reviewer 1 Report

This is an interesting study of HPSIDF in HFD-fed mice. I have following comments for addressing in this study. 

Methods:

-          This study does not include composition of HPSIDF. Is this 100% insoluble dietary fibre? Please include this info.

-          How was the dose of HPSIDF decided? Is 1000 mg/kg referring to 1000/mg/kg body weight/day or 1000 mg/kg feed? In case of 1000 mg/kg of feed, this will be 0.1% of diet. Please indicate if this dose is clinically relevant.

-          Describe the method of euthanasia used in this study.

-          Why was L-carnitine used as a positive control? This is not linked to any results in the discussion.

-          Line 105 – “Weekly and daily recording of weight and food intake.” This seems to be an incomplete sentence.

-          Line 106 – blood collection – “…..taken from the eyeball…..” – Why was retro-orbital bleeding used for blood collection? Were the animals anesthetised during blood withdrawal? Provide sufficient details.

-          Consistently provide centrifugation details. Some places describe the relative centrifugal force and other places have revolutions per minute.

-          Line 113 – which ELISA kits were used? Please provide details of these kits.

-          Line 122 – “Analysis of hepatic fatty acid composition and content” – how was the content measured?

-          Line 153-156: Provide further details of statistical method used for comparison.

Results:

-          Table 1: Is body weight gain presented a correct measure here? The Initial body weight range from 20.02 – 21.45. Will % body weight gain be a better measure to show comparison between the groups?

-          Methods suggests weekly as well as daily food intake measurement. Does table 1 contain weekly or daily food intake?

-          Figure 1D: Can you please include better resolution images for epididymal fat tissue histology? Can you also include adipocyte size from these images?

-          Figure legends should include description for all the abbreviations used.

-          Figure 3B: It is difficult to see steatosis in liver. Can you provide better resolution images?

-          Figure 5: Text in the image is very small and difficult to read. Can you please increase the text size?

-          Figure 6: PC group has been left out from the results included in this figure. Can you please explain why this is the case? Or better to include results from PC group.

Discussion:

-          Line 264 – “…….results from forward-looking cohort studies…..” – Please provide references for these studies.

-          Line 267 – “……excessive concentration of fat…” – Do you mean “…..excessive amount of fat…” here?

-          Some results are not discussed or mentioned at all in the discussion. For example – no comparison has been made with PC group. Why was PC group included?

-          Line 289: What is “shortness reaction”?

-          Lines 294-296: Rewrite the sentence to convey the correct interpretation of your results.

-          Lines 297-299: “It is reported that the oxidative overex-pression of peroxisome fatty acids inhibits mitochondrial fatty acid oxidation, leading to impaired fatty acid oxidation and metabolic disorders.” – No reference(s) are provided for this statement.

References:

-          Numbering should be carefully checked and corrected. After [1-3], the numbering jumps to [48-50] followed by [8-11] in the introduction. Please number the references according to the order of appearance within the text.

-          Very few references cited are from 2018-2023. Please cite recent references.

General comments:

- Moderate English changes required.

- Many abbreviations are presented without defining them at the first use.

Author Response

Insoluble dietary fiber from soybean residue (okara) exerts anti-obesity effects by promoting hepatic mitochondrial fatty acid oxidation

Response to referees

Reviewer #1: 

1.This study does not include composition of HPSIDF. Is this 100% insoluble dietary fibre? Please include this info.

Answer: We have supplemented the base composition of HPSIDF (Table S1). The IDF content of HPSIDF was 90.5% (References 1 as below).

Table S1. The basic composition of HPSIDF

Sample

Protein

Starch

Moisture

Ash

CDF

15.40 ± 1.11 a

3.98 ± 0.78 a

8.30 ± 1.67 a

1.22 ± 1.88 a

HPSIDF

3.12 ± 0.97 b

0 b

3.50 ± 1.83 b

1.12 ± 1.76 a

CDF, crude soybean dietary fiber. Values are expressed as g per 100 g dry matter (mean values). Different letters in the same column (a and b) are significantly different (p < 0.05). The results are expressed as mean ± SD (n = 3).

References

1.Wang, S.; Sun, W.; Swallah, M.S.; Amin, K.; Lyu, B.; Fan, H.; Zhang, Z.; Yu, H. Preparation and Characterization of Soybean Insoluble Dietary Fiber and Its Prebiotic Effect on Dyslipidemia and Hepatic Steatosis in High Fat-Fed C57BL/6J Mice. Food Funct. 2021, 12, 8760-8773, doi:10.1039/d1fo01050f.

  1. How was the dose of HPSIDF decided? Is 1000 mg/kg referring to 1000/mg/kg body weight/day or 1000 mg/kg feed? In case of 1000 mg/kg of feed, this will be 0.1% of diet. Please indicate if this dose is clinically relevant.

Answer: In our previous study (References 1 as below), the optimal dose of HPSIDF (1000/mg/kg body weight/day) was determined by setting up low, medium and high dose groups.

References

  1. Wang, S.; Sun, W.; Swallah, M.S.; Amin, K.; Lyu, B.; Fan, H.; Zhang, Z.; Yu, H. Preparation and Characterization of Soybean Insoluble Dietary Fiber and Its Prebiotic Effect on Dyslipidemia and Hepatic Steatosis in High Fat-Fed C57BL/6J Mice. Food Funct. 2021, 12, 8760-8773, doi:10.1039/d1fo01050f.

  1. Describe the method of euthanasia used in this study.

Answer: We have added the method of euthanasia in lines 111-113 of the revised manuscript. This section has been marked in red.

  1. Why was L-carnitine used as a positive control? This is not linked to any results in the discussion.

Answer: Several studies have shown that L-carnitine can promote fatty acid metabolism in the body (References 1-2 as below). The aim of this study was to investigate the interventional effect of HPSIDF on fatty acid oxidation in high-fat diet-fed mice, so L-carnitine was chosen as a positive control.

References

  1. Alhasaniah, A.H. L-Carnitine: Nutrition, Pathology, and Health Benefits. Saudi J. Biol. Sci.2023, 30, 103555, doi:10.1016/j.sjbs.2022.103555.
  2. Esmail, M.; Anwar, S.; Kandeil, M.; El-Zanaty, A.M.; Abdel-Gabbar, M. Effect of Nigella Sativa, Atorvastatin, or L-Carnitine on High Fat Diet-Induced Obesity in Adult Male Albino Rats. Pharmacother.2021, 141, 111818, doi:10.1016/j.biopha.2021.111818.

  1. “Weekly and daily recording of weight and food intake.” This seems to be an incomplete sentence.

Answer: We have improved this sentence in lines 110-111 of the revised manuscript. This section has been marked in red.

  1. blood collection – “…..taken from the eyeball…..” – Why was retro-orbital bleeding used for blood collection? Were the animals anesthetised during blood withdrawal? Provide sufficient details.

Answer: The method is simple, easy to operate, less harmful to mice, with a high success rate and low mortality (References 1-2 as below). We first anesthetized the mice using ether and then used capillary tubes to collect blood from the posterior orbital venous plexus of the mice.

References

  1. Ruifang Feng, Xiaoxiao Zou, Kai Wang, Huaigao Liu, Hui Hong, Yongkang Luo, Yuqing Tan,

Antifatigue and microbiome reshaping effects of yak bone collagen peptides on Balb/c mice,

Food Bioscience, Volume 52, 2023, 102447, ISSN 2212-4292, https://doi.org/10.1016/j.fbio.2023.102447.

  1. Chunxiao Wang, Zhizhou Chen, Margaret A. Brennan, Jie Wang, Jianfeng Sun, Haibin Fang, Min Kang, Charles S Brennan, Jianlou Mu, The effect of extruded multigrain powder on metabolism and intestinal flora of high-fat-diet induced C57BL/6J mice, Food Research International, Volume 169,

2023, 112878, ISSN 0963-9969, https://doi.org/10.1016/j.foodres.2023.112878.

  1. Consistently provide centrifugation details. Some places describe the relative centrifugal force and other places have revolutions per minute.

Answer: We apologize for not consistently providing centrifugal details. Since centrifugal force and centrifugal speed cannot be directly converted, we have not described this uniformly to ensure the accuracy of the test method.

  1. which ELISA kits were used? Please provide details of these kits.

Answer: We have provided details in lines 118-121 of the revised manuscript. This section has been marked in red.

  1. Analysis of hepatic fatty acid composition and content” – how was the content measured?

Answer: We have added to the quantitative method in lines 149-162 of the revised manuscript. This section has been marked in red.

  1. Provide further details of statistical method used for comparison.

Answer: We have provided the details of the statistical method in lines 175-177 of the revised manuscript. This section has been marked in red.

  1. Table 1: Is body weight gain presented a correct measure here? The Initial body weight range from 20.02 – 21.45. Will % body weight gain be a better measure to show comparison between the groups?

Answer: Since the initial weights of the mice in each group were not identical, a direct comparison of the final weights would not be an accurate indication. Therefore, we chose to use body weight gain to investigate the effect of HPSIDF on the body weight of HFD-fed mice. Recently, there are many studies that have reported using this measure (References 1-3 as below).

References

  1. Feng, K., Lan, Y., Zhu, X., Li, J., Chen, T., Huang, Q., ... & Cao, Y. (2020). Hepatic lipidomics analysis reveals the antiobesity and cholesterol-lowering effects of tangeretin in high-fat diet-fed rats. Journal of agricultural and food chemistry, 68(22), 6142-6153.
  2. Liu, H., Chen, T., Xie, X., Wang, X., Luo, Y., Xu, N., ... & Li, J. (2021). Hepatic lipidomics analysis reveals the ameliorative effects of highland barley β-Glucan on Western diet-induced nonalcoholic fatty liver disease mice. Journal of agricultural and food chemistry, 69(32), 9287-9298.
  3. Sang, T., Guo, C., Guo, D., Wu, J., Wang, Y., Wang, Y., ... & Wang, X. (2021). Suppression of obesity and inflammation by polysaccharide from sporoderm-broken spore of Ganoderma lucidum via gut microbiota regulation. Carbohydrate polymers, 256, 117594.

  1. Methods suggests weekly as well as daily food intake measurement. Does table 1 contain weekly or daily food intake?

Answer: Table 1 contains only the initial and final body weights, as well as the average food intake. Due to individual differences and small fluctuations in the data, we chose to use the mean to show the overall situation.

  1. Figure 1D: Can you please include better resolution images for epididymal fat tissue histology? Can you also include adipocyte size from these images?

Answer: We have made maximum adjustments to the sharpness of the photos.

  1. Figure legends should include description for all the abbreviations used.

Answer: We have added explanations for all the abbreviations involved in the figures.

  1. Figure 3B: It is difficult to see steatosis in liver. Can you provide better resolution images?

Answer: We have made maximum adjustments to the sharpness of the photos.

  1. Figure 5: Text in the image is very small and difficult to read. Can you please increase the text size?

Answer: We have made maximum adjustments to the sharpness of the photos.

  1. Figure 6: PC group has been left out from the results included in this figure. Can you please explain why this is the case? Or better to include results from PC group.

Answer: This study set up L-carnitine as a positive control group, aiming to preliminarily investigate whether HPSIDF has an intervention effect on fatty acid oxidation in mice on a high-fat diet. In addition, the mechanism of L-carnitine intervention on fatty acid metabolism is well known to the public (References 1-2 as below), so we did not involve the PC group in our in-depth investigation of the HPSIDF intervention mechanism.

References

1.Alhasaniah, A.H. L-Carnitine: Nutrition, Pathology, and Health Benefits. Saudi J. Biol. Sci. 2023, 30, 103555, doi:10.1016/j.sjbs.2022.103555.

2.Esmail, M.; Anwar, S.; Kandeil, M.; El-Zanaty, A.M.; Abdel-Gabbar, M. Effect of Nigella Sativa, Atorvastatin, or L-Carnitine on High Fat Diet-Induced Obesity in Adult Male Albino Rats. Biomed. Pharmacother. 2021, 141, 111818, doi:10.1016/j.biopha.2021.111818.

  1. Line 264 – “…….results from forward-looking cohort studies…..” – Please provide references for these studies.

Answer: We have provided references for these studies in lines 291-292 of the revised manuscript. This section has been marked in red.

  1. Line 267 – “……excessive concentration of fat…” – Do you mean “…..excessive amount of fat…” here?

Answer: We have improved this sentence in lines 292-293 of the revised manuscript. This section has been marked in red.

  1. Some results are not discussed or mentioned at all in the discussion. For example – no comparison has been made with PC group. Why was PC group included?

Answer: In the present study, the PC group was set up only to initially investigate the interventional effect of HPSIDF on fatty acid oxidation in the liver of HFD-fed mice. Moreover, the functional properties and applications of L-carnitine have long been explored and discovered. Therefore, it was not overly described in the discussion section.

  1. Line 289: What is “shortness reaction”?

Answer: We have improved this sentence in lines 314-315 of the revised manuscript. This section has been marked in red.

  1. Lines 294-296: Rewrite the sentence to convey the correct interpretation of your results.

Answer: We have modified the description of the results in lines 320-321 of the revised manuscript. This section has been marked in red.

  1. Lines 297-299: “It is reported that the oxidative overexpression of peroxisome fatty acids inhibits mitochondrial fatty acid oxidation, leading to impaired fatty acid oxidation and metabolic disorders.” – No reference(s) are provided for this statement

Answer: We have provided references for this statement in line 326 of the revised manuscript. This section has been marked in red.

  1. Numbering should be carefully checked and corrected. After [1-3], the numbering jumps to [48-50] followed by [8-11] in the introduction. Please number the references according to the order of appearance within the text.

Answer: The numbering of references has been rearranged.

  1. Very few references cited are from 2018-2023. Please cite recent references.

Answer: I apologize for some of the references not being in 2018-2023. Due to the lack of reports on the effects of natural food components on high-fat diet-induced fatty acid oxidation in the liver, we have very limited literature to refer to.

Reviewer 2 Report

This manuscript discusses the effects of insoluble dietary fiber from soybean residue (okara) on obesity via promoting hepatic mitochondrial fatty acid oxidation. The study idea was reasonable. The introduction is well-written, and the conclusion is concise and clear. Some points that need clarification were listed as follows:

1. In the introduction, line number 37 mentions "impairing TG export". Please clarify this abbreviation.

2. In the material section, “HPSIDF was prepared by the previous method [16].” It is acceptable to add a reference, but it is necessary to briefly explain the contents.

2.1. In the material section, “Analysis of hepatic fatty acid composition and content; After the sample was thawed,” what samples were used? The authors must clarify.

2.2. The methods do not explain the purpose of using L-carnitine. 

3. In the results

3.1. In the legend of the table, the author states that “(a, b, c and d) are markedly different”, compared to what group? Explain what each letter means.

3.2. An abbreviation section is needed before the introduction section.

3.3. In Figure 6, the actin should be written β-actin, please correct the spelling.  

3.4. In 3.6. Effects of HPSIDF on the expression levels of hepatic fatty acid β-oxidation-associated proteins in HFD-fed mice, the PC group is missing in the western blot analysis.

4. In the discussion section, “Although most of the suggested beneficial effects of fiber intake are attributed to the viscosity of soluble fiber, results from forward-looking cohort studies concordantly suggest that the intake of insoluble fiber, but not soluble fiber, is strongly linked to the decreased risk of obesity and overweight.” Where are the references for these studies?

Author Response

Insoluble dietary fiber from soybean residue (okara) exerts anti-obesity effects by promoting hepatic mitochondrial fatty acid oxidation

Response to referees

Reviewer #2: 

  1. In the introduction, line number 37 mentions "impairing TG export". Please clarify this abbreviation.

Answer:  We have clarified this abbreviation in line 37 of the revised manuscript. This section has been marked in red. 

  1. In the material section, “HPSIDF was prepared by the previous method [16].” It is acceptable to add a reference, but it is necessary to briefly explain the contents.

Answer: Crude soybean dietary fiber (CDF) with 60% IDF content was purchased from Shandong Sinoglory Health Food Co., Ltd (Liaocheng, China). The CDF was used to prepare the HPSIDF by the enzymatic method. The specific conditions are as follows: α-amylase at 95°C for 35 min, neutral protease at 60°C for 30 min, and amyloglucosidase at 60°C for 30 min.

  1. In the material section, “Analysis of hepatic fatty acid composition and content; After the sample was thawed,” what samples were used? The authors must clarify.

Answer: We have explained this in line 131 of the revised manuscript. This section has been marked in red. 

  1. The methods do not explain the purpose of using L-carnitine.

Answer: We have added the reasons for choosing L-carnitine as a positive control in the methods section in lines 108-110 of the revised manuscript. This section has been marked in red.

  1. In the legend of the table, the author states that “(a, b, c and d) are markedly different”, compared to what group? Explain what each letter means.

Answer: The letters a, b, and c do not have any meaning in themselves, they are compared with each other to show whether the difference is significant or not. We first arrange the means of each group from largest to smallest, and mark the letter a after the largest mean; use this mean to compare with each mean in turn (a downward process), marking the same letter a whenever the difference is not significant, until we encounter a mean with a significant difference, followed by the letter b, and so on down the line.

  1. An abbreviation section is needed before the introduction section.

Answer:we have added the abbreviation section in lines 394 of the revised manuscript. 

  1. In Figure 6, the actin should be written β-actin, please correct the spelling. 

Answer: We have corrected the spelling. 

  1. In 3.6. Effects of HPSIDF on the expression levels of hepatic fatty acid β-oxidation-associated proteins in HFD-fed mice, the PC group is missing in the western blot analysis.

Answer: This study set up L-carnitine as a positive control group, aiming to preliminarily investigate whether HPSIDF has an intervention effect on fatty acid oxidation in mice on a high-fat diet. In addition, the mechanism of L-carnitine intervention on fatty acid metabolism is well known to the public (References 1-2 as below), so we did not involve the PC group in our in-depth investigation of the HPSIDF intervention mechanism.

References

1.Alhasaniah, A.H. L-Carnitine: Nutrition, Pathology, and Health Benefits. Saudi J. Biol. Sci. 2023, 30, 103555, doi:10.1016/j.sjbs.2022.103555.

2.Esmail, M.; Anwar, S.; Kandeil, M.; El-Zanaty, A.M.; Abdel-Gabbar, M. Effect of Nigella Sativa, Atorvastatin, or L-Carnitine on High Fat Diet-Induced Obesity in Adult Male Albino Rats. Biomed. Pharmacother. 2021, 141, 111818, doi:10.1016/j.biopha.2021.111818. 

  1. In the discussion section, “Although most of the suggested beneficial effects of fiber intake are attributed to the viscosity of soluble fiber, results from forward-looking cohort studies concordantly suggest that the intake of insoluble fiber, but not soluble fiber, is strongly linked to the decreased risk of obesity and overweight.” Where are the references for these studies?

Answer: We have provided references for these studies in lines 291-292 of the revised manuscript. This section has been marked in red.

Round 2

Reviewer 1 Report

Thanks for addressing the comments. 

I think some comments still need further attention - 

Please address the following in the revised version.

- Please indicate in the discussion if the selected dose is clinically relevant.

- Please list the agent used for anaesthesia.

- Which ELISA kits were used? You have not provided sufficient details.

- Analysis of hepatic fatty acid composition and content” – how was the content measured? – authors have failed to address this comment. The section 2.6 describes fatty acid composition analysis but how did you measure liver TC, liver TG and liver FFA?

- Abbreviations are only explained in figure 1. Other figures and tables should have similar changes included.

Author Response

Insoluble dietary fiber from soybean residue (okara) exerts anti-obesity effects by promoting hepatic mitochondrial fatty acid oxidation

The response to the reviewer 

Reviewer #1:

Please indicate in the discussion if the selected dose is clinically relevant.

Answer: We added the clinical implications of the selected dose in the discussion section of the revised manuscript (lines 297-299).  This section has been marked in red.

Please list the agent used for anaesthesia.

A: Based on previous studies (references 1-2 below), we anesthetized mice using ether inhalation.

Citation

1.  Isken, F.; Klaus, S.; Osterhoff, M.; Pfeiffer, A.F.H.; Weickert, M.O. Effects of Long-Term Soluble vs. Insoluble Dietary Fiber Intake on High-Fat Diet-Induced Obesity in C57BL/6J Mice. J. Nutr. Biochem. 2010, 21, 278–284, doi:10.1016/j.jnutbio.2008.12.012.

2.  Janovska P, et al. Impairment of adrenergically-regulated thermogenesis in brown fat of obesity-resistant mice is compensated by non-shivering thermogenesis in skeletal muscle. Mol Metab. 2023 Mar; 69:101683. doi: 10.1016/j.molmet.2023.101683.

Which ELISA kits were used? You have not provided sufficient details.

Answer: We provide the ELISA kits used in this study in the 2.1 Materials and reagents section of the revised manuscript (lines 85-89).  Free fatty acids (FFA), 3-hydroxybutyric acid (3-OHB), acetoacetate (ACAC), CPT1, ACOX1, ACS, malonyl-coa, ACC, hydrogen peroxide (H2O2), acetyl-coa, citrate synthase (CS) and succinyl-coa were measured according to the detailed instructions of ELISA kits The concentration of A (SCoA).

Analysis of hepatic fatty acid composition and content” – how was the content measured? – authors have failed to address this comment. The section 2.6 describes fatty acid composition analysis but how did you measure liver TC, liver TG and liver FFA?

A: We have added the calculation of fatty acid content in lines 149 to 163 of the revised manuscript.  This section has been marked in red.  In addition, we describe the determination of hepatic TC, TG, and FFA in Section 2.3.  Specifically, hepatic TC and TG levels were measured using commercial assay kits, and FFA levels were measured using ELISA kits.

Abbreviations are only explained in figure 1. Other figures and tables should have similar changes included.

A: We have explained the abbreviations in the other charts.  Insoluble dietary fiber from soybean residue exerts anti-obesity effects by promoting fatty acid oxidation in liver mitochondria.